# Zinc in Cognitive Impairment and Aging

**DOI:** 10.3390/biom12071000

**Published:** 2022-07-18

**Authors:** Ruize Sun, Jue Wang, Juan Feng, Bin Cao

**Affiliations:** Department of Neurology, Shengjing Hospital of China Medical University, 36 Sanhao St., Shenyang 110004, China; drsunruize@163.com (R.S.); wjw_999@126.com (J.W.); fengjuan99999@hotmail.com (J.F.)

**Keywords:** zinc-associated protein, zinc, zinc finger protein, aging, cognitive impairments

## Abstract

Zinc, an essential micronutrient for life, was first discovered in 1869 and later found to be indispensable for the normal development of plants and for the normal growth of rats and birds. Zinc plays an important role in many physiological and pathological processes in normal mammalian brain development, especially in the development of the central nervous system. Zinc deficiency can lead to neurodegenerative diseases, mental abnormalities, sleep disorders, tumors, vascular diseases, and other pathological conditions, which can cause cognitive impairment and premature aging. This study aimed to review the important effects of zinc and zinc-associated proteins in cognitive impairment and aging, to reveal its molecular mechanism, and to highlight potential interventions for zinc-associated aging and cognitive impairments.

## 1. Introduction

Zinc plays an irreplaceable role in growth and development. Studies have shown that zinc supplementation improved serum insulin-like growth factor 1 (IGF-1) levels and promoted growth in infants born at term, with non-organic growth disorders [1]. IGF-1 is associated with infant height, and IGF-1 deficiency may lead to placental and fetal growth restrictions [2]. A randomized double-blind trial on children aged 4–5 years found that vitamin A supplementation along with zinc increased linear growth in height for age and reduced the risk of infection in children. IGF-1 levels were significantly higher in individuals with concomitant supplementation of vitamin A and zinc [3].

The hormone levels and biochemical indicators in the body gradually decline and deteriorate with age. In particular, the effectiveness of antioxidants in the nervous system decreases during the aging process. An increase in reactive oxygen species (ROS) is considered to be the main pathological process of aging. Cellular damage is followed by significant ROS production. The mortality of neuronal cells in a state of oxidative stress increases, which leads to protein degeneration and neurodegenerative lesions. It has been found that substances such as myostatin can reduce aging-induced β-amyloid (Aβ) plaque formation and deposition in brain regions. Furthermore, myostatin improves the level of cognitive function and has an anti-aging effect, although there was no benefit in young rats [4].

Neurons containing zinc ions can be found in various regions of the brain, including the cortex, amygdala, and hippocampus [5]. The average concentration of zinc ions in the brain is approximately 150 μmol/L, which is 10 times higher than that in serum [6]. Unlike iron and copper, zinc is not redox-active under physiological conditions, and is present in the body in three main forms: free zinc, vesicular zinc, and protein-bound zinc [7]. Most zinc in the brain is in the form of free zinc or non-compactly bound zinc. In addition, it is mostly stored in the vesicles of excitatory glutamatergic synaptic neurons and is released into the synaptic gap along with glutamate upon stimulation of neurons [8]. Due to these properties, zinc plays a key structural role, is involved in catalytic and signaling components, and is found in about 10% of proteins [9]. Zinc-containing proteins are fundamental components of more than 2000 enzymes and proteins in the body, which are involved in the regulations of DNA and RNA synthesis and hormone-receptor interactions, and act as second messengers in intracellular signaling, neurotransduction, neurocytogenesis, and neuronal growth [10]. Zinc is a molecular signal for immune cells, and intracellular zinc homeostasis is maintained by 14 zinc-regulated, iron-regulated transporter-like proteins (ZIP), 10 zinc transporters (ZnT), and other zinc-related proteins, such as metallothionein. All of these zinc-related proteins are irreplaceable in zinc homeostasis. Owing to these sophisticated regulatory pathways, the zinc content of the nervous system can be stabilized within an acceptable range to maintain normal brain function when there is a deficiency or excess of zinc in the body circulation, in a short period of time [3].

Cognitive impairment is usually associated with decreased neuronal apoptosis [11], synaptic dysfunction, a reduction in myelinated neurons and brain volume, and a thinner cerebral cortex. With an increase in age, the number of brain neuron stem cells decreases, which stimulates more calcium ions to enter the neurons to protect neurons and enable cognitive functioning [12]. However, at the same time, aging leads to decreases in glucose transporters in neurons and the level of mitochondrial oxidative phosphorylation and an increase in the production of ROS. This results in oxidative stress damage to neurons, decreases their ATP utilization ability [13,14] and mitochondrial division, and causes impairment in autophagy. These factors cause impaired cognitive functions. Zinc is important for the brain to function normally; however, the serum concentration of zinc gradually decreases with age. Zinc deficiency is closely related to cognitive impairment, memory impairment, and other diseases [15]. The purpose of this article is to summarize the role and mechanism of zinc in aging and cognitive impairment, and to identify new ways to intervene in age-related cognitive impairment.

## 2. Zinc and Neurobiology

### 2.1. Zinc and Neuronal Development

The role of zinc in growth and development has been widely reported for many years. Experimental studies have confirmed its importance in growth and development. Poor nutrition or poor environmental conditions during early growth and development may have irreversible effects on the nervous system. Zinc deficiency is widespread in developing countries. This review will discuss zinc in relation to neurodevelopment, neurogenesis, and cognitive impairment [16].

Inadequate growth in children is closely associated with abnormal neurodevelopment. Nutritional development, especially brain development, is influenced by many factors such as energy, and levels of protein, fat, carbohydrates, copper, iron, selenium, zinc, iodine, vitamins A, B12, C, D, thiamin, folic acid, choline, and long-chain polyunsaturated fatty acids, which are essential nutrients [17]. Zinc, among other minerals, is involved in the composition of many proteins and enzymes in the body. As early as 1986, zinc deficiency was reported to be linked to abnormal brain development as well as learning and memory deficits in young rats [18]. In mid- to late gestation, administration of maternal lipopolysaccharide (LPS) in mice activates an immune response, causing a redistribution of zinc from the fetus to the mother. The fetus develops a zinc deficiency, and astrocytes proliferate in the fetal brain. Oxidative substances and pro-inflammatory cytokines are released, and apoptosis increases, leading to abnormal neurodevelopment [19]. In recent years, it has been found that zinc supplementation throughout pregnancy in LPS-treated rats increased glutamate decarboxylase 67 (GAD67) expression in male pups but did not affect catechol-O-methyltransferase (COMT), while no changes in either GAD67 or COMT expression levels were found in female pups. This study highlighted the importance of zinc supplementation throughout pregnancy, which may limit the adverse effect of maternal infection on fetal neurodevelopment [20].

Zinc, an essential trace metal element in the body, is involved in many cellular signaling and metabolic pathways. *ZNF292* is a gene that encodes a highly conserved zinc finger protein (ZFP). When *ZNF292* is mutated, a range of neurodevelopmental abnormalities may occur, including intellectual disability (ID), autism spectrum disease (ASD), attention deficit-hyperactivity disorder, etc. [21]. High concentrations of zinc are neurotoxic, so zinc levels must be strictly regulated in the body. Prolonged zinc deficiency can have serious adverse effects on the nervous system. Maintaining zinc homeostasis enhances synaptic plasticity. Zinc supplementation also alleviates synaptic defects caused by ASD-related genetic alterations, particularly in the structure and function of excitatory glutamatergic neuronal synapses [22].

The importance of metal ions and their associated receptors on neurodevelopment has been described. However, given the complexity of neurodevelopment, the underlying mechanisms for these effects remain unclear. Neurodevelopment is regulated by a variety of proteins and metal ions. Zinc is involved in the composition of multiple proteins and enzymes in vivo, and as such, plays a key role in neurodevelopment. Zinc homeostasis is maintained through the regulation of ZFPs. Neurodevelopment may require a stable internal environment. Therefore, moderate amounts of zinc supplementation during periods of active neurodevelopment, i.e., during the neonatal period and pregnancy, may be recommended.

### 2.2. Zinc and Neurogenesis

Normal brain development requires the involvement of neural stem cells (NSC). NSC can differentiate into different central nervous system (CNS) cells, including neurons, astrocytes, and oligodendrocytes. Neurogenesis is an important process in the development of the CNS. Neurogenesis can be divided into the following steps: 1. proliferation and renewal of neural stem/progenitor cells (NSE), 2. migration of neurons to different regions of the brain, and 3. differentiation of neurons into different neuronal cell types [23].

These processes generally occur during gestation or the neonatal period. Adult neurogenesis differs from fetal or neonatal neurogenesis. Zinc is believed to be a key factor in the regulation of adult brain stem cell proliferation and neurogenesis [21]. Neurogenesis is considered part of the etiopathology of Alzheimer’s disease (AD). The hippocampus is one of the most dominant areas of neurogenesis in the brain. Neurogenesis, in turn, influences key factors in the development of AD, such as progerin, APP, and amyloid deposition. Therefore, neurogenesis is considered an important component of AD [24]. It has been reported that a decrease in sonic hedgehog signaling in the hippocampus of patients with AD causes a decrease in NSC proliferation and differentiation, which reduces astrocyte proliferation. It also causes a loss of hippocampal stem cells and decreased neuronal production [25]

It has been shown that CXXC5 is an unmethylated CpG dinucleotide-binding protein with two highly conserved zinc finger structural domains. It is involved in cell proliferation driven by E2-ERα signaling [26]. Cell proliferation is the first step in neurogenesis. Zinc plus cyclo-(His-Pro) (CHP)-treated progenitor cell proliferation was significantly increased in the subgranular zone (SGZ) of the hippocampal dentate gyrus (DG). Zinc deficiency decreases NSC proliferation and inhibits NSC development [27]. However, the specific process of adult neurogenesis refers to the generation of new neurons in the mature brain. It has been shown that adult neurogenesis occurs mainly in two sites: the subventricular zone and the hippocampal DG [28] Vesicular zinc is mainly concentrated in the presynaptic terminals of the DG granule cells. Neurogenesis occurs mainly in the SGZ. New hippocampal neurons are continuously generated. Furthermore, many studies have demonstrated that zinc is important in hippocampal neurogenesis after epilepsy and brain injury [29].

Cognitive impairment may occur after paclitaxel treatment, which is known as chemotherapy-induced cognitive impairment (CICI). A mouse model of CICI showed that vesicular zinc and ZnT-3 expression levels decreased after paclitaxel treatment. This negatively affected the differentiation of progenitor cells but did not affect the value-added effects of progenitor cells. When vesicular zinc stores are disrupted, it may further affect one or more stages of neurogenesis, resulting in cognitive impairment [30].

Notably, because zinc is so important, the question has been raised as to whether zinc supplementation would be beneficial to the organism. One study found that in obese mice, who were fed a high-fat diet, low-dose zinc (15 ppm) supplementation improved lipid oxidation and increased levels of neurogenic markers, such as brain-derived neurotrophic factor (BDNF); conversely, high-dose zinc (60 ppm) supplementation decreased levels of neurogenic markers. Moreover, zinc may affect the neural regeneration of hippocampal neurons [31]. This also suggests that zinc supplementation may not be not beneficial for an organism; thus, zinc homeostasis needs to be maintained. ZnT-3, the classic zinc transporter protein, plays an important role in neurogenesis. The number of mature and immature neurons is reduced in mice with *ZnT-3* gene deletion. ZnT-3 protein was found to co-stain along with axon terminals of Brdu + MAP2+ cells in DG neurons. The expression levels of pERK and pCREB were increased by supplementation with ZINC, given as a mixture of zinc chloride and N-acetylcysteine, a glutathione precursor, in a 1:2 M ratio, further suggesting that zinc may activate CREB activity. CREB increases BDNF mRNA expression through activation of ERK1/2, to increase adult hippocampal neurogenesis and cell proliferation. Zinc supplementation in ZnT-3 genetically deficient mice results in enhanced neurogenesis. Whether extracellular zinc is responsible for these effects is still unclear [32].

Neurogenesis is an important stage in neurodevelopment and is implicated in AD. The importance of zinc in neurogenesis further proves its importance in neurobiology. Whether in newborns, children, or adults, old and new neurons are constantly changing. Neurogenesis occurs throughout the lifespan, so its importance cannot be overstated. Zinc supplementation can increase cell proliferation and neurogenesis through multiple pathways. Furthermore, zinc intervenes in the process of neurogenesis to influence the course of neurodegenerative diseases. Therefore, zinc may be a potential therapeutic target for the treatment of neurodegenerative diseases, especially AD.

## 3. Zinc and Aging

Zinc plays an irreplaceable role in chronic diseases related to aging. Human aging is a multifactorial process, which is inseparably linked to oxidative stress and chronic inflammation. Serum zinc levels decline with age but decline more rapidly in patients with AD. Zinc deficiency is more severe in patients with AD, when compared to controls of the same age [33].

### 3.1. Zinc and Hypoxia

Oxidative stress is an underlying factor in many age-related chronic conditions such as atherosclerosis, heart disease, neurodegenerative diseases, and even tumors. Zinc plays a crucial role in antioxidant action during oxidative stress. Zinc inhibits NADPH oxidase and reduces the production of OH^-^, O^2^^-^, and H_2_O_2_ (ROS). Zinc is also a cofactor for the enzyme, superoxide dismutase, which catalyzes the conversion of O^2-^ to H_2_O_2_ [34]. A progressive decline of intracellular ATP levels can be caused by excess zinc, while emodin (6-methyl-1,3,8-trihydroxyanthraquinone) reduces the induction of apoptosis and inhibits cellular ATP production. This could suppress ROS production and endoplasmic reticulum (ER) stress and inactivate the AMPK signaling pathway [35]. Repressor 1 silencing transcription factor (REST), an important transcriptional repressor, was observed in peripheral mononuclear cells among late-onset AD patients. Its minimal promoter region was highly methylated, and a significant reduction in antioxidant enzyme transcription was found. It was demonstrated that REST methylation may induce alterations in AD pathophysiology, which may cause the production of ROS [36].

The relationship between AD, age, and zinc has been studied. Age and sex were found to be risk factors for Aβ accumulation in AD, affecting the regulation of extracellular Zn^2+^ in the brain. Aging-induced brain mitochondrial damage may lead to slow Zn^2+^ reuptake, resulting in approximately four-fold higher extracellular zinc levels in the brain of older animals than in younger adult animals. Zn^2+^-induced soluble Aβ oligomers may produce transient toxicity. In addition, amyloid bodies in AD are zinc-rich, but not strongly associated with cognitive impairment [37]. In the D-galactose-induced senescence mouse model, considerable damage and deterioration in the hippocampal region was observed. The hippocampal damage and deterioration were markedly reduced in the H- intracellular zinc polysaccharides (IZPS) group, indicating that IZPS had more antioxidant and anti-aging capacity [38]. With aging, dopaminergic deficiency may also lead to elevated zinc levels, and a lack of dopaminergic regulation leads to impaired memory [39].

### 3.2. Zinc and Inflammation

Aging is characterized by an impaired immune response and systemic chronic inflammation. Zinc deficiency is an important factor in immune aging. When zinc levels in aged mice are restored through dietary zinc supplementation to a level comparable with young mice, various immune marker levels are improved, including reduced MCP1 and increased naive CD4+ T cell markers. Zinc also reduces activation-induced production of IFNc, interleukin-17, and tissue necrosis factor (TNF)-α cytokines in LNT cells. This suggests that improved zinc status can partially reverse the immune dysfunction and reduce chronic inflammation associated with aging [40]. The NLRP3 inflammatory complex accelerates cognitive dysfunction in AD, and disease progression can be mitigated by zinc supplementation. It was found that in mice deficient in the NLRP3 inflammatory complex, cognitive decline due to zinc deficiency was attenuated [39]. Use of the zinc chelator, *N*, *N*, *N′*, *N′* tetrakis(2-pyridinylmethyl)-1,2-ethanediamine reduces the LPS-induced increase in the formation of NLRP3-level IL-1β, while reducing neuronal death [41]. In addition, nutritional zinc deficiency can increase the levels of the inflammatory factors, CCL2, CXCL2, CXCL5, IL-1, IL-6, and TNF-α in the serum and in colonic mucosa, accelerating the development of colonic tumors [42]. In animal experiments, zinc supplementation of 120 mg/kg improved the growth performance of Pekin ducks; it reduced intestinal damage and increased intestinal enzyme activity. The ability of zinc to inhibit the expression of LPS-induced inflammatory factors and apoptosis-related genes was also demonstrated in in vitro experiments [43].

The prevalence of AD increases with age. The aforementioned studies suggest that AD may be related to oxidative stress, impaired energy metabolism, and chronic inflammatory responses in vivo. Zinc is involved in many of these responses, and zinc plays a key role in a variety of age-related metabolic changes. Future research should investigate how to decrease aging through the modulation of zinc homeostasis or by inhibiting the aforementioned oxidative stress and inflammation pathways.

## 4. Zinc and AD

The prevalence of neurodegenerative diseases is increasing yearly with the advances in modern medical technology and increased human life expectancy. As a representative disease, AD is not yet treatable with specific drugs, and treatment with zinc has been intensely studied. Zinc can regulate the function and metabolism of APP. This affects the formation of plaques and fibrous tangles. It also affects the progression of AD. The hypothesis of AD pathophysiology are concluded in Figure 1.

### 4.1. AD: Background

AD is the most common form of dementia in the elderly. It is a degenerative disease of the CNS characterized by progressive cognitive dysfunction and behavioral impairment that occurs in old age. Clinical manifestations include memory impairment, aphasia, dysfunction, dyscognition, impairment of visuospatial abilities, impairment of abstract thinking and computation, and personality and behavioral changes.

AD can be divided into familial AD (FAD) and sporadic AD. Cases of sporadic AD, i.e., without a clear family history, are more numerous. Mutations in the genes encoding APP, progerin 1, and progerin 2 in FAD have been widely reported [44]. Despite the differences in mutations, the pathological changes in the AD brain share the same characteristics, including Aβ plaques, neurogenic fiber tangles (NFTs), and extensive neuronal and synaptic losses. APP is cleaved by β- and γ-secretase to produce Aβ peptides that aggregate into plaques or fibrous tangles. As expected, AD develops slowly; decreased glucose metabolism or increased Aβ deposition in the brain can occur as early as 10 or even 15 years before the onset of symptoms [45]. Since clinical trials of drugs targeting Aβ reduction alone have not been proven effective, the treatment of AD is still being studied [46,47]. However, researchers have turned their attention to other topics. The apolipoprotein E gene (*ApoE*) has been identified as a major genetic risk for late-onset, sporadic AD. *ApoE* regulates zinc homeostasis [48]. Alterations in trace metal homeostasis may contribute to the development of AD and other neurodegenerative disorders [49]. One pathophysiological hypothesis of AD has included the cholinergic, amyloid, tau, and metal hypotheses, etc. (Figure 1) [49].

The infection hypothesis proposes that the brain tissue becomes infected with *Porphyromonas gingivalis*, which produces gingipains, a class of trypsin-like cysteine proteinases, leading to AD. The LPS-PG in *P.*
*gingivalis* is integrated into the outer membrane vesicles (OMV) and is subsequently transported through the OMV to the CNS, regulating the process of vesicle production and formation. Prolonged exposure to LPS from *Porphyromonas* may induce the accumulation of Aβ (Figure 1).

Furthermore, iron imbalances have also been previously reported: Lf, an iron-binding protein, is upregulated in the brain of patients with AD, which blocks Aβ aggregation, spreads the tau lesion, and causes neuronal damage, resulting in neuroprotective effects [50]. The vascular hypothesis proposes that cerebral hypoperfusion causes chronic hypoxia in the whole brain, and an imbalance in oxidative stress homeostasis. Increased pro-inflammatory factors and ROS damage the blood-brain barrier, reducing the clearance of Aβ and eventually causing white matter damage, synaptic dysfunction, and neurodegenerative lesions. Investigators believe this may be one of the important mechanisms in the development of AD (Figure 1) [51]. Although most studies are in the experimental research stage, important progress has been made. A few studies have already entered clinical trials. Therefore, the future treatment of AD may not be limited to a single route of drug therapy.

### 4.2. APP Processing

The pathological changes in AD are mainly the presence of Aβ plaques and NFT formation, of which Aβ is a key component. Aβ is a product of APP processed by catabolic enzymes. APP is a membrane protein that can be hydrolyzed by two types of enzymes. One involves cleavage by α-secretase in the non-amyloid processing pathway, and the other is cleavage by β-secretase in the amyloid processing pathway. APP is cleaved by α-secretase to form sAPPα and the C-terminal fragment, C83. Conversely, APP is cleaved by β-secretase to form sAPPβ and the C-terminal fragment, C99. The cleaved residues are then cleaved again by γ-secretase to release P3, two major Aβ peptides (Aβ40, Aβ42), and the APP intracellular domain (AICD) (Figure 1). Aβ peptides play important roles in the pathogenesis of AD, with Aβ42 considered harmful [52,53]. Zinc and selenium were found to have neuroprotective effects, probably through their antioxidant properties and inhibitory effects on γ-secretase activity. They reduce the production of Aβ1-40 during APP cleavage processing [54]. Notably, zinc promotes the dimerization and oligomerization of APP-C99 and has an inhibitory effect on γ-secretase. Zinc triggers these effects by binding to substrates at both His-13 and His-14 sites. In contrast, binding of zinc to Lys-28 increases the production of Aβ43 (a precursor of Aβ40) (Figure 2). At different concentrations of zinc, the differences in binding sites lead to different effects. This may also explain the paradoxical role of zinc in AD [55]. Zinc can induce APP trans-dimerization at low concentrations; at low zinc concentrations, C186 and C187 in the E1 structural domain bind to zinc. They have a higher affinity for the appE1 structural domain than the E2 structural domain [56]. The formation of Aβ plaques and NFT after the formation of Aβ peptide aggregation by APP cleavage is the hallmark pathological change of AD. Overexpression of APP is sufficient to cause AD. Recently, APP has been found to be upregulated in astrocytes in an age-dependent fashion. The Alzheimer’s disease β-secretase enzyme, BACE1 inhibitors can alter the autonomic function of astrocytes by reducing amyloid production, providing a new target for future treatment of AD [57]. Recently, it has also been shown that scaffold protein, X11L, is a regulator of APP β-site cleavage. X11L forms a tripartite complex with APP and Alca, and regulates metabolism, affecting APP. When Alca is deficient, increased Aβ production leads to enhanced amyloid plaque formation [58]. The combination of fish oil with selenium and zinc inhibits APP overexpression and alleviates learning and memory deficits in a mouse model of aging [59].

In summary, the etiology of AD is difficult to determine and is associated with many factors, including thyroid and cerebrovascular diseases, and family history. The pathogenesis of AD is complex, and there is no fully effective treatment available. The link between zinc and AD has also been heavily investigated in recent years, with the vast majority of studies referring to the maintenance of zinc homeostasis. Although some progress has been made in zinc therapy, there are still shortcomings. Chronic zinc exposure inhibits recognition and memory [60]. Therefore, whether zinc can be used to treat AD still requires extensive experimental and clinical studies. Thus, it becomes very important to continue investigating the pathogenesis of AD to determine appropriate treatments, including zinc therapy.

### 4.3. Zinc Dyshomeostasis and AD

#### 4.3.1. Zinc in AD

Zinc, copper, iron, calcium, and magnesium were reported to play important roles in AD [61,62]. The zinc homeostasis disorder hypothesis has also been proposed [63]. Zn^2+^ promotes the formation of stable Aβ40 and Aβ42 oligomers. This, in turn, leads to the formation of mature amyloid fibrils. Zn^2+^ is involved in the regulation of synaptic activity, which plays a role in the formation of Aβ oligomers. In turn, synaptic activity plays an important role in the formation of Aβ oligomers [64]. In AD patients, zinc levels are significantly higher in the cortex, hippocampus, and amygdala, and large amounts of Zn^2+^ can be detected in fiber tangles and plaques [65].

Zinc homeostasis is maintained by zinc transporter proteins. The roles of the two families of zinc transport proteins are completely opposed. During the development of AD, there are dynamic changes in the expression of zinc transporter proteins, such as increased ZnT-1 and decreased ZnT-3 expressions. Further control of zinc levels in the brain by modulating expressions of the transporter proteins could be a possible direction for AD treatment [66]. PBT2 acts as a zinc-copper ion carrier. The ability of PBT2 to clear Aβ aggregates in the cortex has been demonstrated in animal studies. Early improvements in cognitive function were observed in phase IIa clinical trials, and the treatment was considered safe [67]. In recent years, it has also been reported that chloroquine (ClioQ) can attenuate the formation of Aβ oligomers in cells [68]. In addition, a protein thought to have similar effects to EDTA, a potent zinc chelator, was found (S100B) to affect Aβ aggregation through two mechanisms; it interacts with fibrotic Aβ42 to decrease amyloid production and may bind competitively to Zn^2+^ to reduce toxic oligomer formation [69].

#### 4.3.2. Zinc Can Be a Double-Edged Sword in AD

It has been shown that 100 μM Zn^2+^ accelerates the protofibrillation of ΔK280 of full-length human tau when compared with 50 μM Zn^2+^. Pathological concentrations of Zn^2+^ significantly enhance tau aggregation-induced apoptosis and toxicity in Cys-291 and Cys-322-dependent SH-SY5Y cells [70]. Zinc binds directly to tau monomers and subsequently stimulates tau protein phosphorylation by activating GSK-3au monomers. Zinc also induces protein phosphatase 2A inactivation and tau hyperphosphorylation through an Src-dependent pathway, ultimately leading to exacerbation of AD-like tau pathology (Figure 2) [71]. Zn^2+^ levels are significantly increased in the cortex and hippocampus of AD mice. Selenoprotein P redistributes the levels of intracranial zinc ions. When selenoprotein P expression increases, the level of synaptic zinc and the expression level of ZnT-3 increase simultaneously. The level of membrane channel protein, NMDAR2A, increases and competes with NMDAR2B for extracellular zinc. This can activate the Src-TrkB or/and BDNF pathways to inhibit tau protein pathological processes and Aβ aggregation in AD mouse models. It also preserves the activity of synaptic proteins, thus improving spatial learning and memory impairment [71]. Confocal immunofluorescence microscopy showed that ZnT-3 expression was distributed along the outside of the GFAP-stained astrocyte cytoplasm, and ZnT-3 localized to the cell membrane of astrocytes. Activated astrocytes, with the help of ZnT-3, internalize Zn^2+^ in dystrophic neurons, thus inhibiting the growth of amyloid plaques. The lack of synaptic Zn^2+^ after Aβ aggregation and plaque deposition may cause clinical symptoms of AD [72]. This also confirms that zinc homeostasis is regulated by zinc transport proteins (e.g., ZnT and ZIP) and metallothioneins (MTs). ZIP proteins transport extracellular and vesicular zinc into the cytoplasm to increase cytoplasmic zinc levels. Conversely, ZnT proteins excrete or remove intracellular zinc by vesicular uptake of zinc [73].

In contrast, increasing zinc levels in the brain can decrease AD progression. Nanoparticles (NPs) encapsulated with zinc were applied to the APP23 mouse model as a noninvasive method to increase brain zinc. It was observed that the plaque area was reduced in the experimental group and that IL-6 and IL-18 levels in the experimental group decreased after treatment. Although the level of IL-10 did not significantly correlate, the overall IL-10 level was increased. These results indicated that there was a decrease in the inflammatory response and Aβ deposition after treatment with zinc [74]. Subacute administration of zinc induces increased expression of CCL2 and CCR2. In addition, FGF2 and IGF-1 overexpression may exert antioxidant and anti-inflammatory effects after the hypoxic-ischemic process. It was demonstrated that subacute zinc treatment had a neuroprotective effect and induced an increase in neuronal plasticity [75]. A study of the genetic background revealed that older adults carrying the ZIP2 Leu2 (Arg43Arg) gene showed a marked reduction in the inflammatory factors Mcp-1, TNF-α, and RANTES after zinc supplementation. Not only did this variant prove to be beneficial for maintaining zinc homeostasis, but the study demonstrated a significant anti-inflammatory effect of zinc [76].

Synaptic zinc was found to be involved in cognitive activity. Both deficiency and excess were associated with cognitive decline [77]. Zinc can exert both neuroprotective and neurotoxic effects and promote the formation of amyloid plaques. Different concentrations or forms of zinc do not have the same pathophysiological effects. Zinc deficiency also exacerbates the rate of cognitive decline. Zinc supplementation decreased indicators of inflammation and reduced plaque deposition. Apoptosis was also significantly enhanced with increasing doses of zinc. Whether zinc is beneficial or harmful in patients with AD should be further investigated. The toxic effects of zinc homeostasis on the pathology associated with AD are shown in Figure 2.

## 5. Zinc and Vascular Dementia

### 5.1. Vascular Dementia

Among the dementia subtypes, vascular dementia (VD) is the second most common [78]. Post-stroke dementia may occur after stroke, especially in patients with a history of multiple strokes. The manifestations of VD are similar to those of dementia, such as memory impairment, decreased executive function, and decreased social skills. VD is usually progressive, and cognitive impairment may become aggravated over time. It has been shown that ischemia induces APP cleavage and enhances deleterious pathways, such as increased conversion of secreted neurotoxic Aβ and the β-secretase pathways [79]. After the onset of ischemia and reperfusion injury, increased levels of zinc-binding proteins release zinc. Zinc homeostasis in hippocampal neurons is disrupted, and large amounts of zinc accumulate in hippocampal neurons, exhibiting neurotoxicity. The use of zinc chelators can reduce the damage to hippocampal neurons [80], and glutathione can reduce neurotoxicity due to zinc accumulation [81].

### 5.2. Risk Factors of VD

Obesity may be a risk factor for the accelerated progression of VD. Compared to VD alone, cognitive impairment in VD with obesity is exacerbated and accompanied by disruption of presynaptic membranes and inhibition of BDNF pathways [82]. In addition to obesity, a history of parental dementia, ApoE variants, diabetes, hypertension, and hypercholesterolemia are all strongly associated with VD. Furthermore, studies have revealed a strong association between *rs9923231*, a gene encoding vitamin K epoxide reductase subunit 1, and VD [83]. Atrial fibrillation may also increase the risk of dementia, including not only VD, but also cognitive impairment and AD [84].

### 5.3. Metal Homeostasis and VD

Metal homeostasis has been found to be important in VD [85]. Following the onset of transient ischemia or stroke, impaired energy metabolism due to abnormal blood supply causes cell membrane depolarization and massive release of glutamate into the synaptic gap. The large amount of glutamate causes overstimulation of its receptors. Currently, known receptors for glutamate include the N-methyl-d-aspartate (NMDA)-type, amino-3-hydroxy-5-methyl-4- isoxazolepropionic acid (AMPA)-type, and erythropoietin-type receptors. Stimulation of the receptors may lead to an imbalance in Ca^2+^ homeostasis in vivo. The entry of Ca^2+^ into glutamate-responsive neurons triggers the death of vulnerable neurons, such as pyramidal neurons in the hippocampus, an area associated with learning and memory [86]. These processes lead to the development of VD. In addition to calcium ions, sublethal concentrations of copper ions (Cu^2+^) significantly exacerbated Zn^2+^-induced neurotoxicity. Cu^2+^ and Zn^2+^ co-administration also significantly increases the expression of genes associated with the endoplasmic reticulum stress response, including *CHOP*, *GADD34*, and *ATF4*. Cu^2+^ and Zn^2+^ synergistically promote neuronal death and influence the pathogenesis of VD [87].

Animal models of VD are also being investigated. In these studies, use of a rat 2-VO modified model found that ErbB3 was neuroprotective and had a key role in neural growth. In this rat model, ErbB3 was downregulated, resulting in a diminished role for both ErbB and PI3K-Akt. Both of these are key regulatory components for the protection of the nervous system [88]. In contrast, sublethal doses of zinc increased the expression levels of proteins associated with the phosphatidylinositol 3 kinase (PI3K) pathway, and decreased the expression levels of proteins associated with neuronal cell death, which reduced apoptosis induced by hypoxia in NSCs via its neuroprotective effect [89]. Recently, it has also been found that myostatin (β-alanylhistidine), an endogenous dipeptide, had antioxidant, anti-crosslinking, and anti-glycation effects. It protected neurons from Zn^2+^-induced neurotoxicity. Myostatin may be a potential drug for the treatment or prevention of VD [90].

As a metal ion, zinc can induce neurotoxicity and induce modulation of the expression of neuroprotection-related channel proteins. The results support the neurotoxic mechanism of zinc, which aggravates VD, AD, and even cognitive impairment. However, some reports suggest the presence of neuroprotective effects of zinc, which can reduce plaque formation [72]. The specific mechanisms and whether zinc is neuroprotective remain unclear. There are few studies on the relationship between zinc and cerebral small vessel disease and the role of zinc in causing VD. Whether zinc can be a therapeutic target or not therefore needs further experimental validation.

## 6. Sleep Disorders

With the accelerated pace of life in modern society, sleep duration and sleep patterns have changed dramatically, and sleep disorders are an increasing concern. Sleep disorders include insomnia, somnolence, REM sleep behavior disorder, restless legs syndrome, periodic limb movements during sleep (PLMS), and obstructive sleep apnea. The most common sleep disorders in AD patients may manifest as frequent daytime naps, difficulty falling asleep at night, nighttime sleep fragmentation, and early morning awakening. Sleep disorders and AD may be co-morbid [91]. Sleep disturbance is a risk factor for dementia and increases peripheral blood Aβ levels [92]. Chronic sleep restriction (CSR) leads to increased accumulation of Aβ42 in the rat brain, especially in the hippocampus and prefrontal lobes. This is accompanied by increased expression of BACE1 and sAPPβ, suggesting that CSR may increase Aβ accumulation through the BACE1 pathway [93]. The formation and accumulation of Aβ is closely related to zinc levels. Zinc is involved in a variety of catalytic enzymes in the formation of Aβ. Zinc is also closely associated with sleep disorders. Insomnia is probably the most prevalent type of sleep disorder in the elderly. In a study, melatonin, zinc, and magnesium supplementation in 43 screened insomnia patients reduced Pittsburgh Sleep Quality Index scores and improved mood and quality of life [94]. Obesity is also an important factor for sleep disorders; regardless of whether they suffer from sleep apnea or insomnia, obese patients with sleep deprivation have low zinc levels, and sleep apnea patients have even lower blood zinc levels [95]. In addition to Aβ accumulation, sleep-wake rhythmic disturbances are clearly associated with tau-mediated neurodegeneration in a bidirectional manner. Tau protein aggregation leads to synaptic dysfunction, eventually leading to loss of synaptic function [96].

PLMS is a repetitive limb movement caused by repetitive muscle contractions in the legs during sleep. PLMS, a sleep disorder, is thought to be associated with cognitive decline, especially in older men with the periodic limb movement index (PLMI) greater than 30, among whom cognitive dysfunction is more pronounced [97].

Similarly, sleep apnea is associated with decreased cognitive function. Female patients with moderate to severe sleep apnea were reported to be at increased risk of cognitive impairment [98].

Sleep disorders may induce abnormal reactions in the body, such as oxidative stress, inflammatory response, abnormal hypothalamic-pituitary-adrenal (HPA) axis activity, and abnormal dopamine secretion [99]. Cognitive impairment due to sleep disorders has also been frequently reported. Sleep disturbances increase Aβ plaques and tau protein aggregation. Whether zinc plays a role in cognitive impairment, which often co-exists with sleep disorders, must be investigated. Several clinical trials have confirmed the benefit of zinc supplementation in sleep disorders. However, further animal and cytological studies are needed to determine the mechanism and whether zinc is the most important trace metal ion affecting sleep disorders and resulting cognitive changes.

## 7. Molecular Mechanism of Zinc-Associated Aging and Cognitive Impairment

Zinc is essential for normal brain function, and there is growing evidence that zinc plays a vital role in age-related cognitive impairment (ARCD). Zinc levels in the brain are tightly controlled, as it is essential for various cellular and molecular processes, enzymatic activities, and transcription factor functions [100]. The concentration of zinc in the brain is 10 times higher than that in the body [7]. While most zinc is bound to proteins or amino acids, approximately 10% of zinc is still chelated or free [101]. Under normal conditions, chelated zinc is located in the vesicles of presynaptic glutamatergic neurons and thus plays a crucial role in regulating neuroplasticity and cognition [102]. After depolarization, approximately 300 μM of zinc is released into the synapse along with glutamate [100,102]. Under healthy conditions, zinc levels are maintained by various zinc transporters and other transporters that balance their export/import into discrete cellular compartments. When these transporters are reduced or dysfunctional, it can lead to severe pathology [103]. Current research suggests that two families of zinc-binding proteins play essential roles in age-related cognitive impairment: zinc transporters (ZnTs) and metallothioneins [1,6,104].

### 7.1. Zinc Transporters

The primary function of ZnTs is to transport zinc out of the cytoplasm or into intracellular organelles [105]. Studies have reported four transcripts of ZnTs in the mammalian brain: ZnT-1, ZnT-3, ZnT-4, and ZnT-6 [105].

#### 7.1.1. Zinc Transporter-1

Zinc transporter-1 (ZnT-1; SLC30A1) is a highly conserved and universally expressed protein [106,107], present on the plasma membrane of neurons and glial cells and is enriched at the postsynaptic density [108,109,110,111]. The expression level and activity of ZnT-1 limit the accumulation of postsynaptic Zn^2+^ after translocation from presynaptic release sites [110,111,112] and are also highly regulated by cytoplasmic Zn^2+^ changes [104,113]; therefore, ZnT-1 plays an essential role in Zn^2+^ homeostasis and protecting neurons from Zn^2+^ toxicity damage. In the presence of acidic extracellular solutions, the cytoplasm acidifies and further accumulates Zn^2+^. ZnT-1 is expressed in the organelles of acidic cells and acts as a Zn^2+^/H^+^ exchanger [104,114], inhibiting ZnT-1-dependent Zn^2+^ outflow in neurons, and as such, enabling the ZnT-1 to provide neuroregulation and neuroprotection function. Researchers have found that ZnT-1 in the hippocampus/parahippocampal gyrus (HPG) in preclinical AD patients was significantly decreased [115], and the same was true in mild cognitive impairment (MCI) patients [105]. As the disease progresses, the production of ZnT-1 increases with increases of intracellular Zn^2+^. Elevated ZnT-1 in late-stage AD (LAD) patients can also lead to increased extracellular zinc, which interacts with Aβ resulting in Aβ aggregation [104,116].

#### 7.1.2. Zinc Transporter-3

Zinc transporter-3 (ZnT-3) is only expressed in the brain and testes, which is the main vesicular zinc transporter in vesicular transporters (VTs). ZnT-3 is located on the membrane of synaptic vesicles (SVs) and transports zinc ions from the cytosol to SVs [117]. Research shows that nearly 36% of SVs carry two VTs, while the SV types containing vesicular glutamate transporter-1 (VGLUT1) and ZnT-3 account for the majority of multi-transporter vesicles (about 34% of all SVs). Co-localization of VGLUT1 and ZnT-3 increases the glutamate content of vesicles, resulting in an enhanced postsynaptic response [118]. The concentration of zinc in SVs is regulated by the activity dependence of glutamate mediated by ZnT-3 [119,120]. Targeted deletion of the ZnT-3 gene also eliminates zinc in SVs, resulting in ARCD and neurodegeneration [121,122]. Zn2þ is a long-term regulator of synaptic plasticity [123]; ZnT-3 mediated synaptic Zn2þ homeostasis is essential for maintaining cognition. In contrast, the loss of ZnT-3 and the resulting synaptic Zn2þ homeostasis disorders may adversely affect memory and cognition. Scientists have found that zinc accumulated in degenerated neurons of ZnT-3-deficient mice [102]. Further research found that after ZnT-3 gene knockout, mice became insensitive to increased Zn^2+^ in the striatum. The electrically induced release of dopamine (DA) was significantly reduced, the clearance rate of DA was higher [124], and ARCD was induced, suggesting that there may be ZnT-3-dependent communication between plasma and neuronal zinc through the cerebrovascular wall [125].

#### 7.1.3. Zinc Transporter-4

ZnT-4 is highly expressed in the brain and mammary gland, while the expression level is low in other tissues. Expression of ZnT-4 decreases significantly with age [126]. ZnT-4 is located on the membrane of intracellular SVs and increases the concentration of zinc in SVs [127], promotes the transport of zinc to endosomes and lysosomes, and protects neurons from zinc poisoning [128,129]. It is reported that the ratio of ZnT-4 mRNA level to neuron-specific enolase mRNA level in the lysosomal system in the brain of post-mortem AD patients increased significantly [130], accelerating the accumulation of Aβ [131,132,133]. Some investigators found that the levels of ZnT-4 in HPG of patients with AD and in LAD patients increased [134]. ZnT-4 is involved in the zinc homeostasis of the cerebral cortex after hypoxia and ischemia. ZnT-4 expression in cerebral microvessels increases, which further damages the blood–brain barrier, thus damaging the neural cells and glial cells related to cognition [135].

#### 7.1.4. Zinc Transporter-6 (ZnT-6)

ZnT-6 isolates cytoplasmic zinc to the TGN and vesicle compartment during the increase of intracellular zinc, destroys the normal sorting and transportation of essential proteins and lipids through the output function of ZnT-6 on plasma membrane or through the secretory pathway, and leads to neuronal degeneration and cytoplasmic inclusion [114]. ZnT-6 is significantly increased in degenerated neurons, and ZnT-6 immunostaining is increased in neurons containing cytoplasmic inclusions in MCI, AD, and Parkinson’s disease (PD). Therefore, it is speculated that ZnT-6 plays a role in the pathogenesis of these lesions [114]. ZnT-6 can promote sphingolipid metabolism by activating smpd1 and converting sphingomyelin into ceramide and choline phosphate [136]. ZnT-6 isolates cytoplasmic zinc to TGN and vesicle compartments during intracellular zinc elevation [137]. ZnT-6 in HPG increased significantly in AD subjects and ZnT-6 increased in patients with MCI, and there was a significant positive correlation between ZnT-6 and Braak score (*p* < 0.10).

### 7.2. Metallothionein

Metallothionein (MT) is a cysteine-rich low molecular weight protein with high metal content. The reactivity and dynamic coordination of MTs with Zn^2+^ are related to the biological requirements of controlling these intracellular metal ions [138]. It can be used to buffer Zn^2+^ and zinc transporters, exporters, and importers. The transient increase of intracellular (cytosolic) Zn^2+^ concentration plays a role in intracellular signal transduction and participates in long-term potentiation [139], which can prevent the damage caused by ROS [140]. The expression of MT is involved in the regulation and control of zinc-dependent cell signal transduction [141]. MT includes four subfamilies: MT-I, MT-II, MT-III, and MT-IV. MT-I and MT-II exist in most tissues, but the expression of MT-III is limited to the CNS, and the expression of MT-IV is limited to stratified squamous epithelium [142].

#### 7.2.1. Metallothionein-I/II

MT-I and MT-II are multivalent proteins. Their main function is to maintain the cellular homeostasis of essential metals such as zinc. They exist in the spinal cord and brain, mainly in the astrocytes [143]. Relevant studies have shown that the expression of MT-I/II was mainly related to the response of astrocytes in the aging brain to oxidative DNA damage in vivo, and the response to oxidative stress in vitro was induced in human astrocytes [144]. MT-I/II gene expression is regulated by antioxidant response elements. Decreased expression of MT-I/II in cell models or knockout animals may increase the sensitivity to oxidative stress [145,146]. Astrocytes and microglia/macrophages around amyloid plaques have high levels of MT-I and MT-II expression in the cortex and hippocampus [147,148], and MT-I, which is mainly upregulated in active astrocytes, has a neuroprotective effect induced by dual mechanisms in AD, namely through Aβ; neurotoxicity is directly and indirectly attenuated by MT-I inhibition of Aβ-induced microglia activation [149]. MT-I and MT-II-deficient mice show impaired spatial learning, but once learned, they retain information like that of wild-type mice [150]. In a study of a semi-PD model in MT-I and MT-II knockout mice, dopamine-doped cells treated with dopamine and L-dopa injection showed that intrinsic MT prevented dopamine quinone-induced neurotoxicity [151]. Scientists have shown that MTs could affect the production of adult neurons and oligodendrocytes. The data showed that MT-I/II played an important role in cell protection and growth regulation during myelin regeneration activated after toxic demyelination injury. In the injured brain, MT-I + II inhibit macrophages, T-lymphocytes, and their IL and TNF-α, matrix metalloproteinases, and ROS formation. In addition, MT-I + II enhances cell cycle progression, mitosis, and cell survival, while inhibiting neuronal apoptosis. MT-I + II is essential for the recovery of neuropathology and cognitive function [152].

#### 7.2.2. Metallothionein-III

MT-III mainly exists in neurons and astrocytes, especially in neurons with high concentrations of zinc, and contains four divalent metal copper and three or four divalent zinc ions [143]. MT-III is not regulated by metals or glucocorticoids, so it is not inducible. Its neuroanatomical distribution is closely related to the region with a high concentration of zinc [147]. MT-III exists in several brain regions at different concentrations, especially in the presynaptic terminals of zinc-rich neurons in the cerebral cortex and hippocampus [148]. MT-III is related to the pathogenesis of neurodegenerative diseases such as PD and AD. It was reported that MT-III was abundantly expressed in the normal human brain but was greatly reduced in the AD brain; in vitro, MT-III inhibits neurite formation and survival of cortical neurons [153,154,155]. In patients with AD, MT-III was identified as a growth inhibitory factor (GIF) or neuronal growth inhibitory factor. MT-III upregulation can be observed in glial cells after brain injury [156]. MT-III has anti-Aβ activity, mainly through the elimination of Aβ. The formation of peptide toxic aggregates to antagonize Aβ plays a role in protecting neurons [157]. The hemiplegic rat model induced by 6-hydroxydopamine (6-OHDA) showed that levodopa treatment significantly increased the expression of MT-III mRNA in the contralateral striatum, suggesting that the regulation of MT-III mRNA expression may be related to the progressive degeneration of PD [158]. MT-III can effectively remove ROS; therefore, exogenous MT-III can prevent neurite extension and differentiated cortical neuron death caused by exposure to high oxygen concentration because it scavenges free radicals more effectively than ZnT-7-MT-I/MT-II at the same concentrations [159]. The above studies show that both ZnTs and MTs play a role in maintaining intracellular and extracellular zinc homeostasis, protecting neurons and regulating growth, and have an important impact on ARCD [Table 1]. Whether there are more zinc-associated proteins involved in this regulatory process still needs further study.

## 8. ZFP and the Cellular Pathways

ZFPs are the largest family of transcription factors in the human genome [160]. They are composed of small domains that bind metal ions (usually zinc) in a tetrahedral geometry using a combination of four cysteine and/or histidine ligands [161,162,163]. In the absence of zinc, these domains expand; in the presence of zinc, the domains of zinc fingers (ZFs) are folded into three-dimensional structures [161]. After one folding, these domains have the function of specifically recognizing DNA, RNA, or other protein targets [9,161,162,163,164,165,166]. ZFs can also directly bind to DNA or RNA and participate in many functions, including transcriptional and translational regulation [167]. Different zinc-binding motifs lead to diversity in the structure and function of ZFs in many human tissues, including the brain [168,169]. The main function of ZFs in the brain is to promote the development of different parts of the brain and the differentiation of neural stem cells. Many studies have shown that ZFPs are closely related to the pathogenesis of nervous system diseases [170]. Researchers found that many ZFP DNA-binding domains downregulated the transcriptional activity of nr2f through missense mutations, to regulate cognitive impairment and optic nerve atrophy [171].

### 8.1. ZFP-1 (CIZ1)

ZFP-1 (CIZ1) is a nuclear protein that interacts with CDKN1A and plays an important role in DNA replication and cell cycle progression [172,173]. Germline or somatic variants in CIZ1 are associated with several neurological and extraneural diseases. Researchers found that after CIZ1 was knocked-down, LPO was upregulated and GSH was downregulated [174], resulting in increased DNA damage, aggravation of neuroinflammation, and aggravation of cell cycle dysfunction [175,176]. Older mice lacking CIZ1 are more likely to acquire age-related cognitive impairment, decreased motor function, and age-related neurological diseases (such as AD). The brain tissue of CIZ1^−/−^ aged mice show significant DNA damage and NF-κB upregulation, oxidative stress, vascular dysfunction, inflammation, and cell death. These harmful effects become more serious in older mice [174]. A variable shear form of CIZ1 was found to be upregulated (CIZ1S) in the hippocampus of AD patients, and the expression of CIZ1S was higher than that of CIZ1 in AD, resulting in the imbalance of the cell cycle [177]. CIZ1 affects defects of the cell cycle and related DNA repair pathways in post-mitotic neurons, which leads to the decline of whole-brain nerve function in the elderly and further aggravates their cognitive impairment [174].

### 8.2. Unkempt

Unkempt is a highly conserved ZF/ring domain protein, which was first found and expressed in the embryonic nervous system of *Drosophila melanogaster.* Researchers found that unkempt played a role in regulating photoreceptor differentiation in the retina during *Drosophila* development in the downstream pathway of a mammalian target of rapamycin (mTOR) [178,179]. Under nutrient-rich conditions, the reduction of unkempt will not affect cell proliferation, but can change the time of photoreceptor differentiation in *Drosophila* [178,180]. Loss of unkempt leads to premature differentiation and pattern defects of photoreceptor neurons in adult eyes [178]. In addition, unkempt is strongly expressed in the larval nervous system of *Drosophila melanogaster* and plays a role in regulating the cell cycle in neural progenitor cells [181]. Mammalian unkempt has the strongest expression in cell lines with a neuronal origin, and its in vivo expression is the strongest in the developing CNS. The expression of unkempt in the developing brain peaks between embryonic days 12 and 18 and is particularly abundant in neurons expressing tuj-1. In vitro experiments show that mammalian unkempt is a conserved ZFP, which regulates mTOR signaling through its ZF domain binding to mRNA, to affect nerve development and function [182,183]. It was reported that inhibition of mTOR signaling by chronic rapamycin treatment enhanced spatial learning, while chronic dietary restriction inhibited mTORC1 to enhance the memory of young mice [184,185]. Both *Drosophila* and mammalian unkempt protein interact with the mTORC1 component of Raptor [186,187]. In the developing nervous system, the expression of mTORC1 component of Raptor decreases in the mice with specific knockout of unkempt, but it does not affect the proliferation of neural progenitor cells and the overall development of the nervous system [180]. However, studies have shown that unkempt was strongly expressed in the cerebellum and hippocampus of adult mice, and unkempt knockout mice recovered their lost learning ability. Since the loss of unkempt increases cognitive flexibility, researchers have linked it to cognitive disorders such as AD [182,188].

### 8.3. Zinc Finger Protein 804A (ZFP804A)

ZFP804A belongs to the ZFP family containing the C2H2 domain. ZFP804A is highly expressed in brain tissue and located in neurites and dendritic spines [189,190,191,192,193]. Researchers found that the expression of ZFP804A in the brain increased in the early fetus [194], peaked at 13−24 weeks after conception, decreased after birth, and further gradually decreased in adulthood [193]. Knockdown of ZFP804A in human NPS altered the expression of genes involved in cell adhesion, mitosis, neural differentiation, and the inflammatory response, suggesting that ZFP804A may be involved in neuronal migration, neurite growth, and synapse formation [195]. Cognitive deficits in many neuropsychiatric disorders are thought to be related to dendritic spines or neurite growth [196,197,198,199]. In the primary cortical neurons of ZFP804A knockdown rats, the neurite growth and the density of dendritic spines are significantly reduced, and the upregulation and downregulation of ZFP804A significantly damage the dendritic morphology; furthermore, different ZFP804A binding proteins show varying levels of reversal of dendritic and synaptic defects [200]. A recent study showed that ZFP804A was necessary for NPC proliferation and neuronal migration, and the overexpression of the gene encoding neurogranin (an SZ risk gene and the target of ZFP804A) can offset the migration defect caused by ZFP804A knockdown [193]. In addition, abnormal ZFP804A expression levels decrease spinal density or abnormal axonal dendration in the cerebral cortex and hippocampus in vivo and in vitro. Female ZFP804A mutant mice show age-dependent defects in sensorimotor gating. Thinking disorder and distraction occurred only after 6 months of age, while the damage to spatial and fear memory was obvious when they were young [201]. Researchers also found that the deletion of ZFP804A had an impact on the language, reading, and cognitive abilities of the population. Therefore, it is speculated that ZFPs are involved in the regulation of cognitive impairment-related diseases through a variety of cellular pathways [202]. In 2q31.2-2q32.3-deletion syndrome, ZFP804A is in the deletion region [203]. Investigators speculate that the absence of ZFP804A may affect 2q31 2-2q32. Patients with this syndrome exhibit impaired reading and spelling [203]. The ZFP804A-rs1344706 allele may be related to better performance of language-related processes [203]. The impairment of ZFP804A-related cognitive functions (word reading, word spelling, non-word reading, and fast naming) constitutes the characteristic pattern of dyslexia [204]. Different zinc-binding motifs have different binding sites and functions in the brain. CIZ1 binds to the nucleus of neurons through the NF-kB pathway and participates in the regulation of the cell cycle and DNA damage. Unkempt participates in the mTOR pathway at the mRNA level, thereby affecting neural development and function. ZF804A is involved in many aspects of neuronal synaptic growth and development. It also has an important impact on human language, reading, and cognitive ability. In conclusion, ZFPs are closely related to nervous system diseases [Table 2], but further cell, animal, and human studies are still required to understand more about the function of ZFPs in the nervous system, to determine whether cognitive impairment-related diseases can be controlled through in vivo and in vitro regulation of ZFPs.

## 9. Zinc and Autophagy

Autophagy is a cellular pathway used to remove and/or recover unnecessary, damaged, and dysfunctional cellular components. Autophagy defects are related to aging-related diseases such as cognitive impairment [205,206]. Zinc plays an indispensable role in apoptosis and cell cycle regulation [207,208]. Convergent evidence from in vivo and in vitro studies shows that zinc is a positive regulator of autophagy [209,210,211,212]. As the number of cells increases and an organism grows, the serum concentration of zinc and autophagy decrease [213,214]. Therefore, the effect of zinc on the autophagy pathway may have great potential as a therapeutic target for aging-related and neurodegenerative diseases such as cognitive impairment [215,216].

### 9.1. Extracellular Signal-Regulated Kinase 1/2 (ERK1/2)

ERK1/2, activated by neurotrophic factors and other chemicals, plays an important role in neuronal differentiation, survival, structural plasticity, long-term enhancement, and memory formation in animal models [217]. The ERK1/2 signaling pathway is associated with many neurodegenerative diseases involving oxidative stress [218]. Abnormal accumulation of activated ERK1/2 occurs in neurons in the brain of patients with AD [219,220]. The mechanism of zinc regulation of autophagy is still unclear, but relevant studies have shown that ERK1/2 plays an indispensable role in regulating autophagy [211,212,221]. Decreased zinc ion concentration in the blood inhibits the activation of the ERK1/2, p38 MAPK, NF-kB, and mTOR pathways [222,223,224]. Zinc can inactivate mTOR and induce autophagy by regulating ERK1/2 and also activate the Beclin 1-PI3K complex through increased phosphorylation of Bcl-2, the negative regulator of Beclin 1 [225]. In addition, zinc can counteract its ERK1/2-mediated phosphorylation by triggering stronger p53 activation [226].

### 9.2. MTs

MT, a low molecular weight protein that binds to zinc, is a potential source of zinc for autophagy activation during oxidative stress [209]. Zinc can induce autophagy gene expression by activating metal reactive transcription factor (MTF1) [227]. The MTs proteins can promote autophagy in a cell cycle-dependent manner through E2F transcription factor 4 (E2F4), a newly reported transcriptional activator of cytoprotective autophagy crucial to zinc homeostasis, to degrade MT proteins, improve the distribution of Zn^2+^ in autophagy, and reduce unstable intracellular zinc ions. Under the condition of oxidative stress [228], the expressions of MtnA and other metallothioneins, genes involved in glutathione metabolism, and other genes known to belong to oxidative stress responses or general stress response, increase. MtnA also responds to genes and pathways of additional metabolic processes, autophagy, and apoptosis [229]. Overexpression of MT in cells can protect lysosomes from estrogen-induced damage and maintain viability while inhibiting autophagy [230]. Knockdown of metallothionein-3 inhibits late autophagy in irradiated glioma cells [231].

### 9.3. Methylation

Gene methylation by N6-methyladenosine (m6A) is the most abundant in mRNA. Zinc can also regulate autophagy by regulating gene methylation [232]. The human n6-methyltransferase complex (MTC) contains a ZF motif sufficient to bind RNA. However, methylation activity requires two ZFPs [233]. YTH domain family protein 3 (YTHDF3) is an important reader in m6A methylation. It plays an important role in regulating the translation of m6A-modified mRNA and affects the decay of methylated mRNA in an m6A methylation-dependent manner [234,235]. Researchers found that ZF E-Box-binding homeobox 1 (ZEB1) is the key downstream target of YTHDF3. YTHDF3 enhances the stability of ZEB1 mRNA in an m6A-dependent manner and positively regulates cell migration, invasion, and the EMT [236]. Arabidopsis HAKAI is necessary for complete m6A methylation, and its knockout leads to a 35% reduction in m6A abundance [237]. ZFPs 1 and 2 (HIZ1 and HIZ2) interact with HAKAI and are members of the m6A writer complex. Their knockdown and knockout cause strong developmental defects [238,239,240].

### 9.4. Zinc Homeostasis

Autophagy decreases with age [241]. The dysregulation of zinc metabolism observed in the elderly may be a result of slower autophagy. Similarly, in other age-related cognitive impairment diseases associated with autophagy deficiency, such as AD, autophagy deficiency may be associated with dysregulation of zinc homeostasis [205]. Although zinc is a positive regulator and protector of autophagy, when the zinc concentration exceeds a certain threshold, excessive zinc leads to uncontrolled autophagy and cell death [205,212]. Relevant studies have shown that high concentration of zinc in the culture medium (1000 μM) can stimulate autophagy-driven apoptosis of porcine small intestinal epithelial cells (IPEC-J2) [242]. Zinc ionophores clioquinol and PCI-5002 induce autophagic cell death in mouse astrocytes and non-small cell lung cancer cells, respectively [243,244]. Interestingly, excess zinc is crucial in activating toxic autophagy induced by tamoxifen and H_2_O_2_ [205,213,245,246]. After spinal cord injury, the serum zinc concentration decreases, the spinal cord zinc concentration increases at the injured site, and the zinc level in infiltrated monocytes increases significantly, which is equivalent to the injured spinal cord, and then zinc infiltrates the lesion area with monocytes [247]. Due to zinc chelation in senile plaques, the local zinc ion dynamics near amyloid aggregates in AD are unbalanced [74]. After long-term injection of zinc nanoparticles into MCAO mice for 2 weeks, the plaque size is significantly reduced, and levels of pro-inflammatory cytokines IL-6 and IL-18 are altered. Whether the autophagy function of nerve cells plays a role is closely related to age-related diseases such as cognitive impairment. Zinc can affect autophagy through ERK1/2, MTs, methylation, zinc homeostasis, and other pathways; however, the specific mechanism of zinc regulating autophagy is still unclear. We still need to further explore the molecular mechanism and cellular pathway of zinc participation in the regulation of autophagy in the nervous system.

## 10. Interventions for Zinc-Associated Aging and Cognitive Impairment

Zinc affects learning and memory and is an important factor in ARCD because it regulates various cellular and molecular processes [100,248]. For example, during brain depolarization [101,124], zinc is released into synapses together with glutamate to regulate synaptic communication and neural plasticity [249].

### 10.1. Clioquinol

Chloroiodoquine is a weak zinc chelator (kD ≈ 10^−7^). Some researchers have found that chloroiodoquine reduced amyloid protein in APP TG2576 mice (AD model). The weak zinc chelation property of chloroiodohydroquine may be the basis of an anti-amyloid effect. Genetic removal of synaptic zinc in the brain of APP TG2576 mice can greatly reduce the plaque load [250], which is similar to the plaque reduction effect of membrane-permeable metal chelator DP-109 [251]. Chloroiodohydroquine can also act as a zinc ion carrier rather than a zinc chelator in physiological solution [252]. Owing to low intracellular free zinc levels, chloroiodohydroquine transfers zinc downward and creates a concentration gradient [253]. Under physiological conditions, chloroiodohydroquine may carry zinc from the extracellular to the intracellular space. This effect may increase in the brain of AD patients, in which Aβ plaques provide more extracellular zinc stores, which may lead to lower intracellular zinc levels [254]. ZnT-3 can load zinc into presynaptic vesicles of the hippocampus; the expression of ZnT-3 is downregulated in accelerated aging mice [255,256,257]. Decreased ZnT-3 protein levels and impaired memory were observed in the cortex of ZnT-3 knockout mice, aged mice, and aged humans [125,258]. Researchers found that chloroiodohydroquine acted as a zinc ion carrier under physiological conditions [243], increasing the intracellular zinc level in the cytoplasm and autophagic vesicles in an extracellular zinc-dependent manner, to restore the cognitive ability of ZnT-3 KO mice [125,259]. In addition, PBT2 (prana Biotechnology), an 8-hydroxyquinolone, can improve synaptic plasticity and cognitive function by mediating different cellular pathways (especially involving glutamatergic signal transduction at synapses), improve protein (CamKII, CREB, AMPA, NMDA, and VGLUT1) redistribution of zinc in the brain, and improve the content of total zinc in the hippocampus of elderly subjects [260]. Investigators found that AD-like transgenic mice treated with PBT2 had improved cognitive function and reduced disease pathology [210]. In a short-term phase IIa human clinical trial of AD, it was found that PBT2 played a positive role in improving cognition and restoring normal brain function [260,261,262].

### 10.2. Rapamycin

Rapamycin is a well-known mTOR inhibitor, which plays an important role in autophagy and insulin signaling [263,264]. The mTOR is a serine/threonine kinase, which plays a key role in regulating protein synthesis and degradation, age-dependent cognitive decline, and the pathogenesis of AD [265,266]. Abnormal mTOR signaling in the brain can affect several pathways related to metabolism, insulin signaling, protein aggregation, mitochondrial function, and oxidative stress in AD [267]. Increased mTOR expression co-localizes with NFT and mediates tau phosphorylation [265,266,268,269]. Investigators found that intracerebroventricular injection of zinc sulfate induced continuous mTOR (s2448)-P70S6K (t389) phosphorylation in rat hippocampus. Rapamycin can improve zinc-induced tau hyperphosphorylation, oxidative stress injury, and synaptic injury by downregulating mTOR/P70S6K activity, upregulating Nrf2/HO-1 activity, and ameliorating spatial learning defects and age-related cognitive impairments [10].

### 10.3. Zinc Supplements

Zinc has antioxidant and anti-inflammatory properties. Zinc supplementation within the limits of a normal diet can alleviate cognitive impairment [10]. Melatonin (Mel) is a neurohormone released by the pineal gland. In healthy individuals, Mel levels peak at night. They regulate the circadian rhythm [270], immune system, and blood pressure [271]. They have strong antioxidant properties and neuroprotective effects and act through the promotion of Aβ to alleviate Aβ-induced neurotoxicity [272]. Researchers found that in physical and mental activities, zinc and melatonin protected against AlCl3-induced AD and related liver and kidney injury by activating antioxidant and anti-inflammatory processes and reducing apoptosis. GSK-3β dysregulation is related to the pathogenesis of a variety of diseases affecting the brain, including neuroinflammatory and neurodegenerative diseases [273]. Wnt signaling plays an important role in maintaining synaptic plasticity and memory in the brain [274]. AlCl3-induced neurodegeneration and p-GSK-3β protein expression were significantly increased, followed by a decrease in the level of Wnt and β-catenin. The combination of zinc and Mel therapy can provide additional neuroprotective benefits, because zinc deficiency leads to the downregulation of the Wnt/β-catenin signaling pathway, and Mel activates Wnt/β-catenin signaling to prevent neuronal cell death [275]. In addition, long-term moderate physical activity can regulate the classical Wnt/β-catenin signaling pathway, thereby improving neuroprotection and synaptic plasticity.

Zinc deficiency is closely related to the pathogenesis and process of ARCD. The above-described studies confirm that it can intervene in ARCD by regulating the way zinc participates in the regulation of the nervous system. However, current research is still limited to known cellular pathways, and we still need to explore more methods to treat zinc-related ARCD.

## Figures and Tables

**Figure 1 biomolecules-12-01000-f001:**
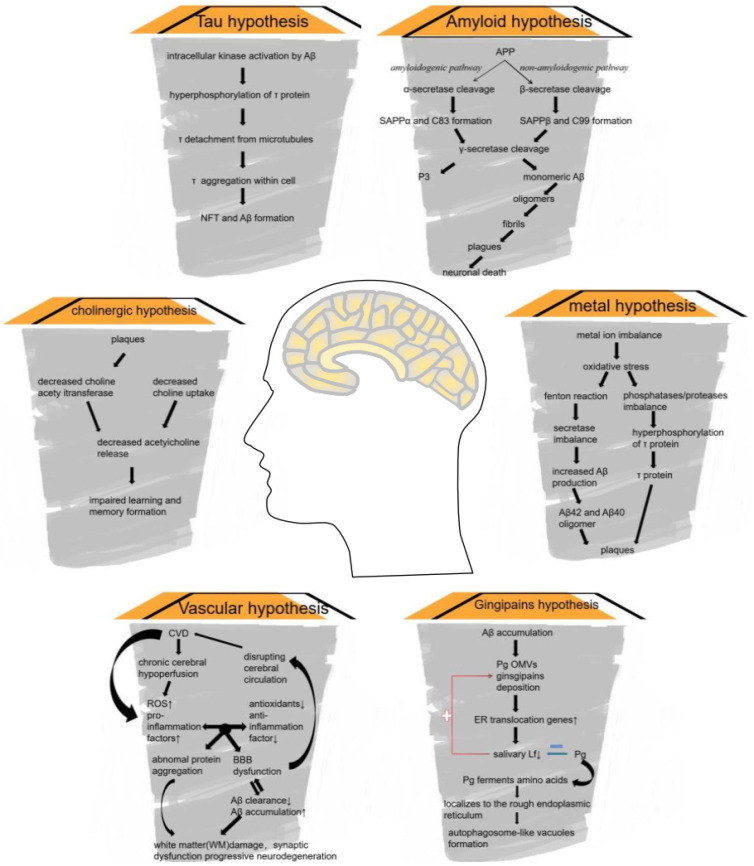
Multiple hypotheses are involved in Alzheimer’s disease (AD) pathophysiology.

**Figure 2 biomolecules-12-01000-f002:**
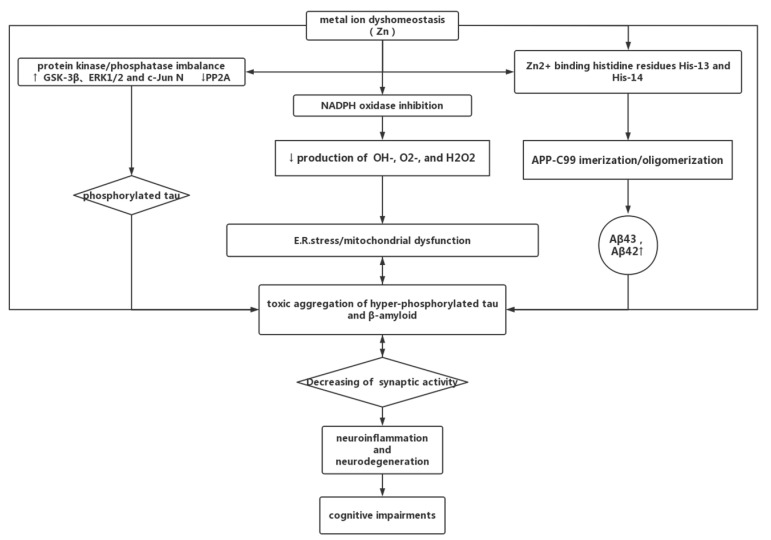
The toxic effects of zinc homeostasis on the pathology associated with AD. Zinc has multiple levels of toxic effects, including oxidative stress and toxic accumulation. Ultimately, cognitive dysfunction is the result.

**Table 1 biomolecules-12-01000-t001:** The molecular mechanism and function of ZnTs and MTs in ARCD.

Zinc Associated Protein		Function	Reference
Zinc transporters	ZnT-1	Limit the accumulation of postsynaptic Zn^2+^ after translocation from presynaptic release sites Protect neurons from Zn^2+^ toxicity As a Zn^2+/^H+ exchanger in acidic cells, it inhibits ZnT-1 dependent Zn^2+^ outflow in neurons ZnT-1 is elevated in AD patients, making Aβ gather	[104,110,112,113,114,115]
	ZnT-3	Transport Zn^2+^ from cytoplasm to SVs and regulate the concentration of Zn^2+^ in SVs Co-localization with VGLUT1 increased the content of glutamate in VTs and the postsynaptic response Maintain Zn2þ homeostasis in synapses, so as to maintain human cognitive function Plasma and neurons depend on it to exchange through the blood vessel wall	[117,118,119,120,123,124,125]
	ZnT-4	Increase the concentration of Zn^2+^ in SVs, promote the transport of Zn^2+^ to endosomes and lysosomes, and protect neurons from zinc poisoning Accelerate the accumulation of Aβ in the brain of patients with AD Further damage the BBB in patients with is, and damage the neural cells and glial cells related to cognition	[127,128,129,131,132,133,135]
	ZnT-6	It separates Zn^2+^ in the cytoplasm into TGN and VTs in cells with high Zn concentration It plays a transport or secretory function on the plasma membrane, destroys the normal sorting and transport of essential proteins and lipids, and leads to neuronal degeneration and cytoplasmic inclusion Activate smpd1 to promote sphingolipid metabolism and convert sphingomyelin into ceramide and choline phosphate	[114,136,137]
Metallothionein	MT-I/MT-II	Maintain intracellular homeostasis of essential metal ions in the human body Induced by oxidative stress in human astrocytes In AD patients, MT-I directly attenuates the neurotoxicity of Aβ and indirectly inhibits Aβ-induced microglia activation and subsequent neurotoxicity Prevention of dopamine quinone induced neurotoxicity in patients with Parkinson’s disease It plays the role of cell protection and growth regulation in the process of myelin regeneration activated after toxic demyelination injury Inhibit macrophages, T lymphocytes, and their formation of interleukin and TNF-α, matrix metalloproteinases, and ROS Enhance cell cycle progression, mitosis, and cell survival, and inhibit neuronal apoptosis	[143,144,145,146,149,151,152]
	MT-III	Inhibition of neurite formation and survival of cortical neurons It has anti-Aβ activity and eliminates the toxic aggregates of Aβ peptide, so as to antagonize the neurotoxic effect of Aβ peptide It can effectively clear ROS and prevent neurite extension and differentiated cortical neuron death caused by exposure to high oxygen concentration	[153,154,155,159]

**Table 2 biomolecules-12-01000-t002:** Cellular pathways involved in ZFP regulation in the nervous system.

Zinc Finger Protein	Function	Reference
CIZ1	Maintain the balance of LPO and GSH and participate in NF-κB, oxidative stress, vascular dysfunction, inflammation, and cell death Affect the cell cycle and related DNA repair pathways of post-mitotic neurons It is involved in regulating ARCD, motor decline, and age-related neurological diseases	[174,175,176]
Unkempt	Regulation of retinal photoreceptor differentiation in the mTOR downstream pathway Regulate the cell cycle of neural progenitor cells Affect cognitive flexibility and participate in the regulation of cognitive impairment diseases	[178,179,181,188,189]
ZFP804A	It affects the expression of genes such as cell adhesion, mitosis, neural differentiation, and inflammatory response, and participates in neuronal migration, axon growth, and synaptic formation Damage dendritic morphology and participate in the regulation of dendritic synaptic defects Affect the spinal cord density or axon dendritic state of the cerebral cortex and hippocampus Regulate people’s language, reading, and cognitive abilities, and participate in the regulation of cognitive impairment-related diseases	[193,195,200,201,202]

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
