# Peer review of "Zinc in Cognitive Impairment and Aging"

_biomolecules, 2022, doi:10.3390/biom12071000_

Round 1

Reviewer 1 Report

In the present study the Author aimed to review the important effects of zinc and zinc-associated protein in cognitive impairment and aging, reveal its molecular mechanism and the probable intervention for zinc associated aging and cognitive impairment. 
Overall, I found this review very interesting, timely, well written and scientifically sound. I have only some minor suggestions aimed to improve the quality of the paper and these are outlined below:

1) I suggest the Authors to write the aims and scopes of the Review in the Introduction section. As well I suggest to briefly discuss cognitive impairment and aging mechanisms in this section.

2) How literature searches were conducted and relevant papers included? Please, add this part to the article.

3) I suggest to carefully check the manuscript with the help of a native English speaker.

Author Response

Dear Reviewer,

Thanks for your constructive comments. These comments are all valuable and helpful for improving our manuscript entitled “Zinc in cognitive impairment and aging”, as well as the important guiding significance to our researchers.

We have studied the comments carefully and have made a correction which we hope to meet with approval. Revised portions are marked in yellow in the revised manuscript. The summary of corrections and the responses to the reviewer’s comments are listed below

Reviewer #1:

  1. I suggest the Authors to write the aims and scopes of the Review in the Introduction section. As well I suggest to briefly discuss cognitive impairment and aging mechanisms in this section.

Thank you for your comment. In the introduction, we have added a brief introduction to the current mechanisms of aging and cognitive impairment, as well as the aims and scopes of this review.

  1. How literature searches were conducted and relevant papers included? Please, add this part to the article.

Thank you for your question. We searched the literatures mentioned in the manuscript through typing in the key words “zinc”, “cognitive impairment”, “aging” in the website of “Pubmed”.

  1. I suggest to carefully check the manuscript with the help of a native English speaker.

We apologize for the poor language of our manuscript. We worked on the manuscript for a long time and the repeated addition and removal of sentences and sections obviously led to poor readability. We have now worked on both language and readability and have also involved native English speakers for language corrections. We really hope that the flow and language level have been substantially improved.

Reviewer 2 Report

biomolecules-1777457-peer-review-v1

In their manuscript, Sun et al. reviewed the role of zinc ions for n cognitive impairment and aging. Specifically, the authors have focused on molecular mechanisms and the probable intervention for zinc-associated aging and cognitive impairment. This review establishes a very broad understanding of how zinc ion plays a pivotal role in the biological context.

Major comments

1.     While the authors extensively describe the aspects of zinc in the text, there is usually a lack of conclusion in each section. Please state the conclusion at the end of each section. This helps readers to understand the effect of zinc on the biological process, discussed.

2.     On the same note, the article lacks figures/images consisting of schematics and the structure of Zn-bound proteins. For example, a schematic showing the ion homeostasis for AD, VD are highly recommended. Also, I think authors should incorporate a few figures demonstrating the structure of Zn-bound proteins as well.

3.     All figure legends need more details. The figure needs to be referenced in the main text

4.     Figure quality is very poor and the writing within the figure is illegible. The figures need to be re-drawn and quality needs to be improved.

Minor comments:

1.     Line 46: reference is missing.

2.     Line 618, 622: references can be merged.

3.     Line 785. there is a syntax error in reporting the Kd value. Also, Kd value needs a unit.

4.     The authors need to double-check the referencing style for this journal. Also, the current style is not consistent. what does this sign “[J]” mean for the references 1-94?

Author Response

Dear Reviewer,

Thanks for your constructive comments. These comments are all valuable and helpful for improving our manuscript entitled “Zinc in cognitive impairment and aging”, as well as the important guiding significance to our researchers.

We have studied the comments carefully and have made a correction which we hope to meet with approval. Revised portions are marked in yellow in the revised manuscript. The summary of corrections and the responses to the reviewer’s comments are listed below

Reviewer #2:

  1. While the authors extensively describe the aspects of zinc in the text, there is usually a lack of conclusion in each section. Please state the conclusion at the end of each section. This helps readers to understand the effect of zinc on the biological process, discussed.

Your suggestion is valuable. At the end of each part, we add the conclusion of the related mechanism of zinc, so that readers can better understand the content of the article. We have briefly added the conclusion below in last paragraph of each part, As well as some parts give future conjectures.

  1. On the same note, the article lacks figures/images consisting of schematics and the structure of Zn-bound proteins. For example, a schematic showing the ion homeostasis for AD, VD are highly recommended. Also, I think authors should incorporate a few figures demonstrating the structure of Zn-bound proteins as well.

I summarize the mechanism of Fe, Cu, Zn, and Al plasma in AD and VD formation, and edit the clear schematic diagram of AD ion steady state at Line 385. Since there is little evidence for Zn and VD studies, partly overlapping with the mechanism of AD. Therefore a separate schematic of the ion steady state for VD is not shown.

  1. All figure legends need more details. The figure needs to be referenced in the main text.

Thank you for your suggestion and we have revised it.

  1. Figure quality is very poor and the writing within the figure is illegible. The figures need to be re-drawn and quality needs to be improved.

Thank you for your suggestion. The images were modified and re-edited and the English writing were corrected by native English speakers.

Round 2

Reviewer 2 Report

biomolecules-1777457-peer-review-v2

 I believe that the authors have tried their best to revise the manuscript as per the suggestion. Unfortunately, it is very hard for me to understand the revisions that the authors have made. I think this manuscript is UNPROFESSIONALLY revised.

1.     There are different highlighted colors in the revised manuscript. There is NO indication in the cover letter regarding the meaning of the DIFFERENT highlightS in the text.

2.     Moreover, I feel, it is completely needless to submit a revised manuscript in a track-changed format.

3.     Between pages 7-9, several paragraphs are merged. It looks authors did not bother to proof check the manuscript before submission.

4.     There are TWO “figure 1” . Figure legend is NOT revised

5.     Figure 2 is illegible. There is no reference to figure 2 in the main text

Author Response

Dear Reviewer,

Thanks for your constructive comments. These comments are all valuable and helpful for improving our manuscript entitled “Zinc in cognitive impairment and aging”, as well as the important guiding significance to our researchers.

We have studied the comments carefully and have made a correction which we hope to meet with approval. Revised portions are marked in yellow in the revised manuscript. The summary of corrections and the responses to the reviewer’s comments are listed below. And we have re-polished the English writing, which can be seen in the “track-changed” format of the manuscript.

  1. I'm sorry for the color anomaly. Because of the color difference in the computer software, the color of the modified text is not consistent. This caused the overall color deviation. We have unified the highlighting color of the modified part. It is convenient for experts to review.
  2. Thank you very much for your suggestion. The track-changed format is used to facilitate expert review of the changes. The editor also recommends using this format for submissions. For your review, we have submitted two manuscripts, one of which is in non-track-changed format.
  3. I am sorry for my carelessness. Due to the modification of the size of Figure 1, some formatting of the article was adjusted. Some text positions were changed when the language was modified. This time we proofread the manuscript carefully.
  4. The track-changed format may have caused the deleted Figure 1 to appear in the text. We have adjusted and re-uploaded the manuscript in non-track-changed format.
  5. Thank you very much for your suggestion, we have read a lot of literature and have some understanding of the homeostasis of Cu, Fe, Al. Due to the word limit, it was not possible to include all of them in the text.Therefore, Figure 2 has been modified to describe mainly the important role of zinc steady state.

Yours sincerely,

Bin Cao